# Automatic Classification of Foot Thermograms Using Machine Learning Techniques

**Vítor Filipe [1,2,\*]** , **Pedro Teixeira [1]** and **Ana Teixeira [1,3]**

1. School of Science and Technology, University of Trás-os-Montes e Alto Douro, Quinta de Prados, 5001-801 Vila Real, Portugal; miguelbento1997@hotmail.com (P.T.); ateixeir@utad.pt (A.T.)
2. INESC TEC—INESC Technology and Science, 4200-465 Porto, Portugal
3. Mathematics Centre CMAT, Pole CMAT—UTAD, Quinta de Prados, 5001-801 Vila Real, Portugal
* Correspondence: vfilipe@utad.pt

**Abstract:** Diabetic foot is one of the main complications observed in diabetic patients; it is associated with the development of foot ulcers and can lead to amputation. In order to diagnose these complications, specialists have to analyze several factors. To aid their decisions and help prevent mistakes, the resort to computer-assisted diagnostic systems using artificial intelligence techniques is gradually increasing. In this paper, two different models for the classification of thermograms of the feet of diabetic and healthy individuals are proposed and compared. In order to detect and classify abnormal changes in the plantar temperature, machine learning algorithms are used in both models. In the first model, the foot thermograms are classified into four classes: healthy and three categories for diabetics. The second model has two stages: in the first stage, the foot is classified as belonging to a diabetic or healthy individual, while, in the second stage, a classification refinement is conducted, classifying diabetic foot into three classes of progressive severity. The results show that both proposed models proved to be efficient, allowing us to classify a foot thermogram as belonging to a healthy or diabetic individual, with the diabetic ones divided into three classes; however, when compared, Model 2 outperforms Model 1 and allows for a better performance classification concerning the healthy category and the first class of diabetic individuals. These results demonstrate that the proposed methodology can be a tool to aid medical diagnosis.

**Keywords:** diabetic foot; thermography; machine learning; classification; clustering; computer vision





## 1. Introduction

Technological advances in infrared cameras improved the speed of response and resolution, which allowed for the application of this technology to different areas, namely, to medical use [1]. With this technique, it is possible to visualize the temperature distribution of the human body and detect thermal variations associated with some diseases, such as breast cancer, vascular disorders, muscle pain, complications related to diabetic foot, among others [1,2].

The application of infrared thermography has increased significantly over the years, especially with regard to the study of problems related to diabetic foot [3,4], that is, one of the main complications observed in Diabetes Mellitus (DM) patients, who have a 12% to 25% risk of developing foot ulcers during life [5,6].

Currently, medical information systems that support detection and decision making are changing, evolving from a traditional manual data analysis to more modern and specialized computer methods, known as Computer-Assisted Diagnosis (CAD), thus providing tools that allow health professionals to achieve more accurate diagnosis [7].

The evaluation of medical images by humans is susceptible to errors due to several factors, namely: negligence, fatigue, and even sensory overload, as consequence of the large amount of information that needs to be analyzed. Other factors to be considered

are that human visual capacities contain limitations, resulting from the visual perception of optical illusions, which affects the accuracy of the diagnosis. It also happens that health institutions may not have enough specialists to perform the diagnostic task [8]. Therefore, it is important to bridge the gap between the user's perception and the accuracy of the diagnosis; as the literature review shows, the development of CAD systems helps to overcome these problems, with image processing algorithms with greater detection performance based on automated segmentation, image enhancement, and restoration and feature extraction and classification approach [3,4,9].

There are several studies that focused on the diagnosis of diabetic foot diseases, using thermal images, based on the measurement of skin temperature variation. Many of them seek to classify foot thermograms through asymmetric analysis, which consists of comparing the temperature of the foot with the contralateral one, in order to define a limit that allows for categorizing between healthy and diabetic [10–12]. This type of approach has shown good results in several studies; however, it has some limitations, for example, when the patient has similar complications in both feet, as it is not possible to detect the risk areas and incorrectly classify the thermogram. Nevertheless, as techniques to detect areas of risk without contralateral analysis were still needed, as well as to have a consistent accurate algorithm that can effectively detect, analyze and classify the minimum thermal variations in the diabetic foot, machine learning techniques have gained an increasing impact in medical areas. The support vector machine algorithm was used by Saminathan et al. [13] to apply asymmetric analysis to foot thermograms and classify them as normal or prone to ulcer.

To combat the limitations previously mentioned, in [14] the authors propose a thermal change index, which represents the degree of variation of the plantar temperature, to group the thermograms of diabetic individuals in five classes; for that, a public plantar thermogram database [15] was used. This public dataset has been used by several researchers with the aim of analyzing the distribution of the temperature of the feet of (non)diabetic individuals. One of them was Cruz Vega et al. [9], who proposed a deep learning technique to classify the thermograms of the feet of diabetic individuals into five classes, by comparing two classes at a time. Another group was Khandakar et al. [16], who proposed a machine learning-based scoring technique with feature selection, optimization techniques and learning classifiers to perform a binary classification of the thermograms of the feet into diabetic and nondiabetic classes. The same investigators also use the public dataset [15] to propose a machine-learning framework, based on a thermal change index, to classify the thermograms of diabetic individuals into five classes [17]. Isaza and Diaz [18] also used the dataset in [15] to propose the implementation of deep learning architectures under data augmentation methods, using the Fourier transform, to classify each foot thermogram separately as belonging to a diabetic or nondiabetic individual. Despite the great progress, there is still a need to develop techniques to classify thermal images that not only present a robust, accurate and detailed classification of the thermograms, but that are also practical enough to be used by health professionals in their clinical practices.

With this in mind, with this work we aim to develop a functional methodology to analyze and classify a diversity of thermal changes in the plantar region of diabetic and healthy individuals; the public dataset [15] was used. Additionally, based on the experience acquired when working with the data from the aforementioned dataset and on the information collected during the literature review, a three-level scale was adopted to classify the feet of a DM individual into three categories. Thus, in this paper, two models, using different machine learning algorithms, for classifying thermographic images of the feet are proposed. In the first one, the foot thermograms are classified into four classes: healthy and three categories for diabetics. In the second, the classification process has two stages: Initially, each foot is classified as belonging to a DM or healthy individual. Then, if the foot is considered an individual DM foot, a classification refinement is made in a second stage.

In order to extract the image descriptors used for foot classification, a cluster algorithm divides the thermogram into different regions. For each region, a set of regional descriptors

are calculated, being used as features to train and test both models. Five machine learning classification algorithms were evaluated in both models: logistic regression (LR), support vector machine (SVM) with two variants, and K-nearest neighbor (k-NN), with two variants.

This paper is organized as follows: In Section 2 the used dataset is introduced and the proposed methodology is described. In Section 3 the obtained results are presented. Finally, Section 4 discusses the results and presents conclusions.

## 2. Materials and Methods

### 2.1. Thermographic Image Dataset

In our study, we used a public dataset of thermal images (thermograms) of the plantar region, available online.

In their work, Contreras et al. [15] described the acquisition protocol to collect the thermograms of both feet of 167 individuals (122 diabetics and 45 healthy) recruited from four medical institutions located in the city of Puebla, Mexico, over a period of 3 years. Before the image acquisition procedure, each participant had a resting period of 15 min, in a room with controlled temperature, in order to reach a state of thermodynamic equilibrium and to improve the accuracy of temperature variation detection. Two different infrared cameras were used for thermogram acquisition (FLIR E60 and FLIR E6). The emissivity parameter was set to 0.98, which is the recommended emissivity for measurement of human skin with an IR camera.

The camera was fixed by a vertical tripod placed one meter away from the feet. The raw thermogram dimensions are $320 \times 240$ pixels, and each one has a corresponding temperature file of equal dimensions, in which the temperature value of each point (pixel in the thermogram) is stored.

After acquisition, the raw images were processed to separate the left and right foot in two different sub-images. These sub-images are segmented and spatial transformed to position the foot in a vertical alignment. The publicly available dataset is composed of individual thermograms of each foot, stored in a *csv* format.

### 2.2. Methodology

With the aim of detecting and classifying abnormal changes in the plantar temperature, a methodology having two processing stages (Figure 1), feature extraction and classification, is proposed.

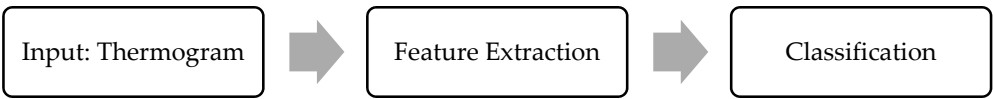

**Figure 1.** Methodology overview.

The first stage consists of the extraction of relevant features from the thermograms. After applying machine learning classification algorithms, the subject's foot is classified as healthy or in one of the three classes of pathology severity.

### 2.3. Feature Extraction

The extraction of features is one of the most important steps in computer-aided diagnostic systems. It intends to extract the most significant and non-redundant information from the images to facilitate the learning process.

Previous studies showed that an increase in temperature in the plantar region of diabetic patients is related with foot ulcer appearance [19]. Based on this fact, we calculate the global temperature features average, maximum and minimum from the entire plantar region to retain the general foot condition. However, these parameters are not enough to characterize the status of the foot, because it does not have a uniform temperature. Regional features should also be extracted to describe the pattern of temperatures in the different sections of plantar region. Following a previous work of the same authors, the K-means

algorithm is applied to divide the foot into five different clusters with different values of temperature [20]. The clustering algorithm allows the distribution of image pixels between the different thermal clusters, as shown in Figure 2.

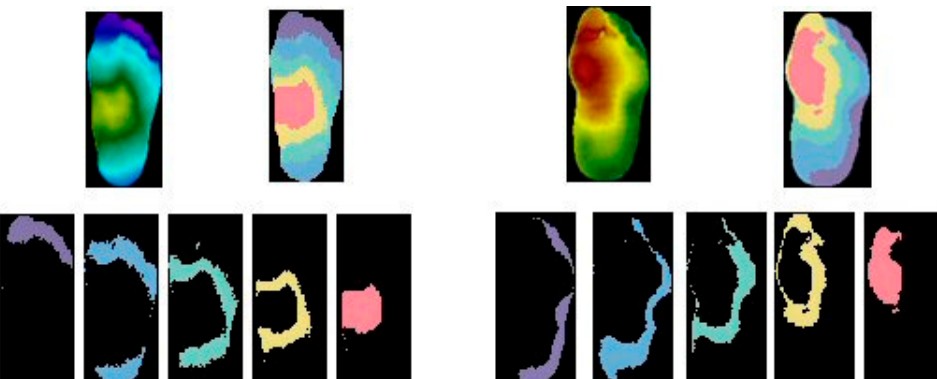

**Figure 2.** Thermogram division in clusters. **First row:** RGB representation of thermal image and the clusters image representation, in non-diabetic (**left**) and diabetic (**right**); **Second row:** the 5 individual clusters.

For each cluster, the average, maximum and minimum temperatures are calculated, obtaining this way a total of 15 regional features of plantar temperatures. The clustering algorithm allows for the detection of small hot spots revealing plantar regions that are prone to the appearance of ulcers. In addition to the features already presented, the *Cluster Thermal Index* (CTI) is also included in the list of features to provide a quantitative estimation of the thermal changes of the foot caused by DM. The CTI provides a measure of temperature deviation between a subject and the control group, taking into account not only the temperature difference between the clusters but also the range of temperature in the control group. The index is based on the average temperature of each cluster in relation to reference temperatures obtained from the healthy individuals (control group). To establish the reference temperatures in the control group, firstly the clusters are sorted in ascending order according with the centroid value. After, the average temperature of each cluster ($\overline{CG_i}$) is calculated. Table 1 shows the reference temperatures obtained in the control group, with 45 subjects.

**Table 1.** Average and standard deviation of the temperature (°C) for the Control Group, per cluster.

| Cluster | 1 | 2 | 3 | 4 | 5 |
|---|---|---|---|---|---|
| $\overline{Tc}$ | 23.93 | 25.52 | 26.59 | 27.61 | 28.88 |
| *Std. dev.* | 1.76 | 1.54 | 1.57 | 1.51 | 0.92 |

The CTI is calculated as the average of the positive differences between the average temperature in each cluster of a subject and the correspondent reference temperatures, obtained in the control group, as is (1).

$$\text{CTI} = \frac{\sum\limits_{i=1}^{k} |CGI_i - IND_i|}{k},$$ (1)

where $k$ represents the number of clusters, ($CG_i$) is the reference temperature of cluster $i$ and $IND_i$ is the average temperature in cluster $i$ for the subject.

Table 2 illustrates the result of feature extraction in thermograms of a non-diabetic and a diabetic subject. A total of 19 features are extracted from each thermogram, which are used to feed the classifiers algorithms of the next stage. It is observed that the global temperature values in the diabetic subject and their respective cluster temperatures are

further away from the reference values, as shown in Table 1. This results in a CTI value of 3.23. In the case of the non-diabetic subject, we observe smaller deviations between the features values and the references temperatures, resulting in an CTI value close to 0 (0.24). The extraction of features in the different clusters allows for the detection of temperature deviations in foot sections that may not be evident when analyzing the global features.

**Table 2.** Extraction of features in the thermograms of two subjects: Non-diabetic and Diabetic.

| Features | | | Non-Diabetic | Diabetic |
|---|---|---|---|---|
| **Global** | | Avg. | 26.75 | 29.75 |
| | | Max. | 29.90 | 33.10 |
| | | Min. | 22.94 | 26.35 |
| **Regional** | Cluster 1 | Avg. | 23.83 | 27.85 |
| | | Max. | 24.80 | 28.23 |
| | | Min. | 22.94 | 26.35 |
| | Cluster 2 | Avg. | 25.82 | 28.67 |
| | | Max. | 26.32 | 28.24 |
| | | Min. | 24.81 | 29.14 |
| | Cluster 3 | Avg. | 26.82 | 29.66 |
| | | Max. | 27.37 | 29.16 |
| | | Min. | 26.33 | 30.18 |
| | Cluster 4 | Avg. | 27.94 | 30.73 |
| | | Max. | 28.55 | 30.20 |
| | | Min. | 27.38 | 31.21 |
| | Cluster 5 | Avg. | 29.13 | 31.76 |
| | | Max. | 29.90 | 31.23 |
| | | Min. | 28.55 | 33.10 |
| **Deviation from reference** | Cluster Thermal Index | | 0.24 | 3.23 |

### 2.4. Classification Algorithms

Image classification involves the extraction of information from an image and then associating it with a class label. Classifying images in the machine learning domain can be approached as a supervised learning task. In this type of learning, the system must "learn" inductively an expression of a model that describes the data, called the target function. This function is used to predict the value of a variable, called a dependent variable or output variable, from a set of variables, called independent variables or input variables, also known with features.

There are several supervised machine learning algorithms that deal with classification. In this study, Logistic Regression, Support Vector Machines and k-Nearest Neighbors are used.

In general, the logistic regression classifier uses a linear combination with more than one input variable as an argument for the sigmoid function, obtaining a value between 0 and 1 as the output of the function. In particular, an input that produces a result greater than 0.5 is considered to belong to class 1. On the other hand, if the output is less than 0.5, the corresponding input will be classified as belonging to class 0.

In the SVM algorithm, the objective is to find a hyperplane in an N-dimensional space (N represents the number of input variables), which classifies the data. In order to separate the classes, there may be several possible hyperplanes; however, the ideal is to find a plane that maximizes the distances between the class data, called the maximum margin. Maximizing the distance from the margin provides some reinforcement so that the points of future data can be classified with more confidence [21]. The classifier has different kernel functions, allowing it to adjust to different types of data. In this study the classifier will be tested with the linear and the quadratic kernel.

The k-NN algorithm consists of assigning a new example for the most common class among its closest k-neighbors. To classify a point $x$ as belonging to one of the classes, the $k$

observed data points closest to *x* are found. That is, the classification rule is to assign *x* to the population that has the most observed data points out of the *k*- nearest neighbors [22]. For example, if *k* is 1, then *x* is simply assigned to your nearest neighbor's class. This classifier allows for changing the number of neighbors (*k*), in order to improve the classification. In this work, 3-NN and Weighted k-NN will be applied.

### 2.5. Proposed Models

In this work, two models for classifying the thermographic images are proposed. The LR algorithm and the variants of SVMs, k-NN were applied to both models.

In the first model, the foot thermograms are classified into four classes: healthy and three categories for diabetics, which represent progressive severity of the disease.

In the second model, the classification consists of two stages: initially a binary classification is performed and the foot thermogram is cataloged as belonging to a healthy or diabetic individual; then, and only for cases classified as diabetic, a second classifier is used to obtain a refinement of the classification for diabetic cases.

### 2.6. Performance Evaluation

To evaluate and compare the performance of classification algorithms, several experiments were carried out using the metrics: Accuracy, Sensitivity, Specificity, Precision, F-score, and Area Under the Curve (AUC). The first five metrics are defined by (2) to (6), where *TP*, *FP*, *TN* and *FN* represent the number of cases of True Positive, False Positive, True Negative and False Negative, respectively.

$$Accuracy = \frac{TP + FN}{TP + TN + FP + FN} \tag{2}$$

$$Sensitivity = \frac{TP}{TP + FN} \tag{3}$$

$$Specificity = \frac{TN}{FP + TN} \tag{4}$$

$$Precision = \frac{TP}{TP + FP} \tag{5}$$

$$F - score = \frac{2 * TP}{2 \times TP + FP + FN} \tag{6}$$

A Receiver Operating Characteristic (ROC) curve was used to capture the trade-off between the true positive rate (sensitivity) and false positive rate (1-specificity) at thresholds over a continuous range.

The AUC summarizes the information obtained using the ROC curve and, instead of depending on a specific operational point, it is an effective and combined measure of sensitivity and specificity that describes the inherent validity of diagnostic tests [23], with an AUC of 1.0 representing a perfect test, while an AUC of 0.5 defines a random test. The higher this metric (closer to 1), the better the discriminatory power of the model.

In binary classification, accuracy, sensitivity, specificity, precision, and F-score are calculated using the expressions (2) to (6). In the case of multi-class classification, those metrics can be calculated for each class in a one-vs.-all method and determine how often each class makes accurate predictions.

As in the used dataset, the classes are not balanced [15], that is, the percentage of samples in each one of the classes is not similar; it is important to pay attention to the impact of class imbalance on the performance metrics of the classification. With this in mind, the metrics that should be taken into greater consideration are F-score and AUC.

## 3. Results and Discussion

This section presents the results of experiments with the two models described in Section 2. In Section 3.2, the results of the multi-classification concerning to Model 1 are described, while in Section 3.3 the results of Model 2 are analyzed.

### 3.1. Experiments

In order to find the features that allow one to obtain better performance, the different classification algorithms were tested and evaluated in different subsets of features. The subset that obtains the best results contains the features referring to the average, maximum, and minimum of the general thermogram of the foot and the respective five clusters. The CTI index is also a relevant resource, so 19 features will be used as inputs to train the different classification algorithms.

To assess the generalizability and validation of the both models, the *k*-fold cross-validation technique was adopted. In this technique, the dataset $D$ is randomly divided into $k$ subsets, $D_1; D_2; \ldots ; D_k$, of approximately equal size. The model is trained and tested $k$ times and, for each time $I = 1, 2, \cdots, k$, it is trained in all subsets, except $Di$, which is used for testing. In this work, the 10-fold cross-validation technique is used to divide the 80% of the whole sample into 10 equal-sized subsamples. While preparing the folds, the percentage of samples for each class was maintained in every fold, named the *stratified-fold* technique. This way, the model gets equally distributed data for training and test folds.

After finding the best classifier algorithm for each one of the two models presented in Section 2.4 and in order to be able to compare them, the remaining 20% of the dataset not used to train/validate the models was used for testing.

### 3.2. Classification with Model 1

In this subsection, the results of Model 1 are presented. In Model 1, the thermogram is classified into one of four classes, healthy and three different categories for diabetics, according to the degree of variation of the plantar regions, with Class 1 being an earlier stage of the disease revealing small thermal variations and Class 3 a more advanced stage of the disease with large thermal variations. These categories are based on the ones proposed in [14], where the authors use the *Thermal Change Index* (TCI), which represents the degree of change in plantar temperature, to group the thermograms into five classes. Class 1 corresponds to a $TCI \leq 2$, Class 2 to a $2 < TCI \leq 3$, Class 3 to a $3 < TCI \leq 4$, Class 4 to a $4 < TCI \leq 5$ and Class 5 to a $TCI > 5$. An example of this classification can be seen in Figure 3.

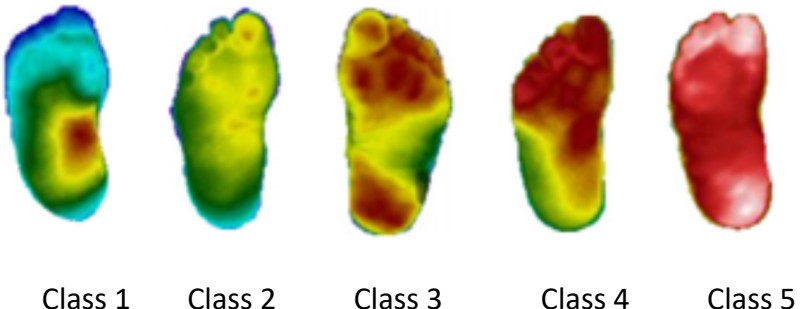

Class 1　　　Class 2　　　Class 3　　　Class 4　　　Class 5

**Figure 3.** Classification of 5 thermograms of diabetic individuals [14].

Given the low number of samples for each class, when compared to the number of healthy examples, in the present work the five categories were regrouped into three. Additionally, as the classification algorithms used are more robust and accurate when applied to balanced datasets, when performing this reduction in the number of classes, a more balanced distribution of the thermograms per class was obtained.

The performed class reduction consists of joining the categories with the closest TCI indexes, thus grouping the feet that have more similar temperatures. Applying this methodology, three classes were defined: Class 1 for $TCI \leq 3$, Class 2 for $3 < TCI \leq 5$

and Class 3 for TCI > 5. Figure 4 presents the reclassification of the thermograms of Figure 3. Thus, considering the new categorization, the dataset used will have 90 healthy thermograms (27% of the total sample), and 244 diabetic ones, divided as follows: Class 1 has 92 thermograms (28%), Class 2 has 64 (19%) and Class 3 has 88 (27%).

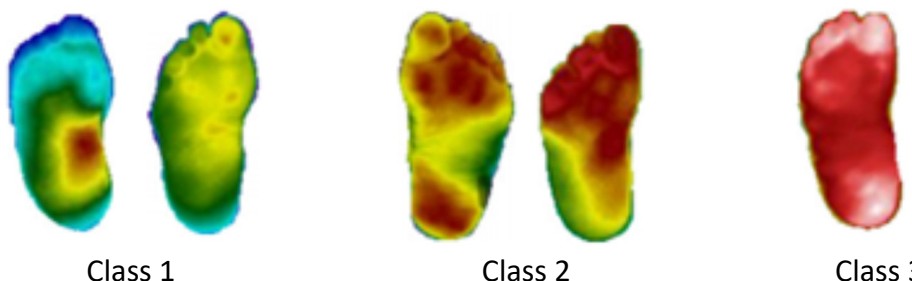

Class 1            Class 2            Class 3

**Figure 4.** Reclassification of thermograms of diabetics in 3 classes.

The evaluation of each classification algorithm is measured, separately, for each one of the four classes, using a one-vs.-all method. The results of the performance obtained for each class with the different algorithms are presented in Table 3. The comparison of the classification algorithms is based on the average values, of the four classes, for each one of the metrics used; these values are presented in the *Average* row in Table 3.

**Table 3.** Metrics scores for Model 1, with 4 classes.

| Algorithm | Class | Accuracy | Sensitivity | Specificity | Precision | AUC | F-Score |
|---|---|---|---|---|---|---|---|
| | Healthy | 0.851 | 0.653 | 0.923 | 0.758 | 0.927 | 0.701 |
| | 1 | 0.843 | 0.797 | 0.861 | 0.686 | 0.923 | 0.738 |
| SVM Linear | 2 | 0.937 | 0.833 | 0.959 | 0.816 | 0.939 | 0.825 |
| | 3 | 0.981 | 0.946 | 0.995 | 0.986 | 0.984 | 0.966 |
| | **Average** | 0.903 | 0.807 | 0.935 | 0.812 | 0.944 | 0.807 |
| | Healthy | 0.854 | 0.778 | 0.883 | 0.709 | 0.915 | 0.742 |
| SVM | 1 | 0.877 | 0.730 | 0.933 | 0.806 | 0.930 | 0.766 |
| Quadratic | 2 | 0.937 | 0.833 | 0.959 | 0.816 | 0.973 | 0.825 |
| | 3 | 0.981 | 0.959 | 0.990 | 0.973 | 0.983 | 0.966 |
| | **Average** | 0.912 | 0.825 | 0.941 | 0.826 | 0.950 | 0.825 |
| | Healthy | 0.817 | 0.708 | 0.857 | 0.646 | 0.891 | 0.675 |
| | 1 | 0.832 | 0.649 | 0.902 | 0.716 | 0.866 | 0.681 |
| 3-NN | 2 | 0.937 | 0.813 | 0.964 | 0.830 | 0.952 | 0.821 |
| | 3 | 0.974 | 0.959 | 0.979 | 0.947 | 0.989 | 0.953 |
| | **Average** | 0.890 | 0.782 | 0.926 | 0.785 | 0.924 | 0.783 |
| | Healthy | 0.817 | 0.708 | 0.857 | 0.646 | 0.907 | 0.675 |
| Weighted | 1 | 0.832 | 0.662 | 0.897 | 0.710 | 0.906 | 0.685 |
| k-NN | 2 | 0.937 | 0.792 | 0.968 | 0.844 | 0.968 | 0.817 |
| | 3 | 0.981 | 0.973 | 0.985 | 0.960 | 0.989 | 0.966 |
| | **Average** | 0.892 | 0.784 | 0.927 | 0.790 | 0.943 | 0.786 |
| | Healthy | 0.866 | 0.764 | 0.903 | 0.743 | 0.924 | 0.753 |
| | 1 | 0.877 | 0.757 | 0.923 | 0.789 | 0.939 | 0.772 |
| LR | 2 | 0.918 | 0.813 | 0.941 | 0.750 | 0.964 | 0.780 |
| | 3 | 0.959 | 0.905 | 0.979 | 0.944 | 0.994 | 0.924 |
| | **Average** | 0.905 | 0.810 | 0.937 | 0.806 | 0.955 | 0.807 |

Although the classes considered are more balanced than the ones in [14], the thermograms are not equally divided between the classes; thus, special attention should be paid to the F-score and AUC metrics.

Analyzing all the results obtained with the different algorithms, presented in Table 3, and focusing, in particular, on the *Average* row, both SVM Linear and LR algorithms present

similar metric results, obtaining a difference lower than 1% between each metric, and the same happens with the 3-NN and Weighted k-NN algorithms; however, the first group performs slightly better than the second group, mainly in the AUC metric, which is almost 2% higher. The SVM Quadratic is the algorithm that presents the best results, obtaining in some metrics a difference of 4% when compared to the algorithm that presents worse metric results.

In Figure 5, the ROC curves obtained with the Quadratic algorithm, for each class, are illustrated.

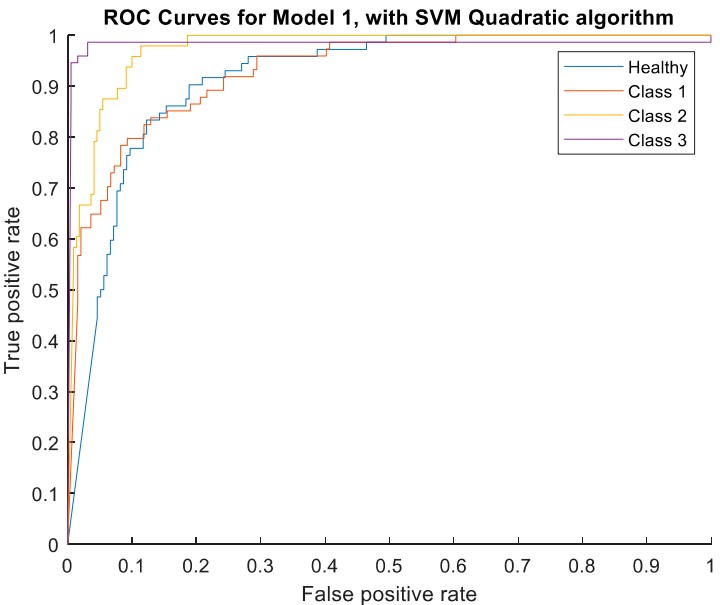

**Figure 5.** ROC Curves for Model 1, with SVM Quadratic algorithm.

From Figure 5, it is perceptible that, although all the AUC results are higher than 91%, the classes with the best classification are Classes 2 and 3, with the two highest AUC values.

### 3.3. Classification with Model 2

In this subsection, the results of Model 2 are presented. This model is proposed in order to improve the performance of the classification of the healthy and diabetic classes; a very important step as it contributes to the division of the classification between healthy and diabetic thermograms.

In Model 2, the classification is divided in two stages: initially a binary classification is performed, in order to catalog the foot thermogram as belonging to a healthy or a diabetic individual; for the feet classified as diabetic, a second classifier is used to obtain a refinement of the classification.

The evaluation of the classification algorithms is measured considering as a true positive when the model correctly predicts the diabetic individual and as a true negative when the model correctly predicts the healthy individual. The performances of the different algorithms are shown in Table 4.

**Table 4.** Metrics scores for stage 1 in Model 2—binary classification.

| Algorithm | Accuracy | Sensitivity | Specificity | Precision | AUC | F-Score |
|---|---|---|---|---|---|---|
| SVM Linear | 0.858 | 0.898 | 0.750 | 0.907 | 0.902 | 0.903 |
| SVM Quadratic | 0.858 | 0.888 | 0.778 | 0.916 | 0.908 | 0.902 |
| 3-NN | 0.787 | 0.832 | 0.667 | 0.872 | 0.872 | 0.851 |
| Weighted k-NN | 0.825 | 0.867 | 0.708 | 0.890 | 0.920 | 0.879 |
| LR | 0.892 | 0.923 | 0.806 | 0.928 | 0.954 | 0.926 |

It can be seen that LR is the algorithm with the highest values in the different metrics, obtaining differences greater than 10% compared to the algorithm 3-NN, which has the worst results, while the SVM algorithms are the second-best ones, with similar results, followed by Weighted k-NN; however, this was higher than the 3-NN algorithm. In Figure 6, the ROC curves for all of the five algorithms are illustrated.

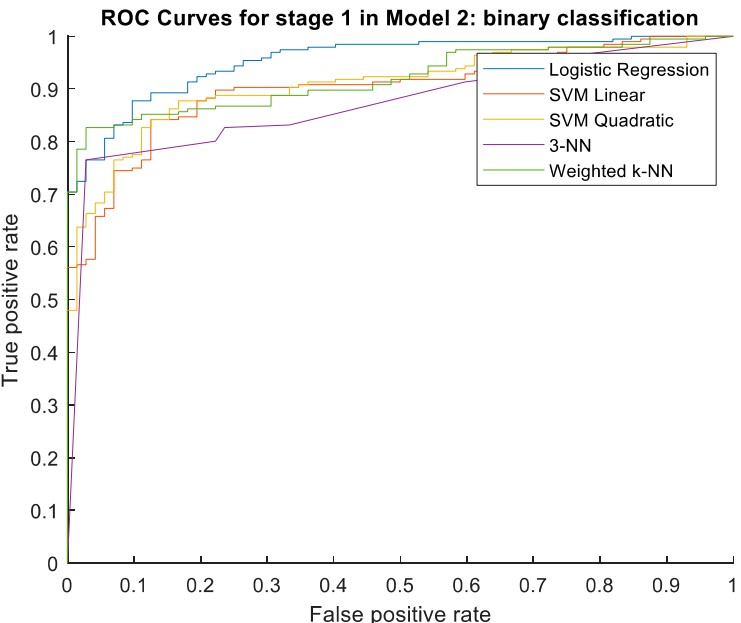

**Figure 6.** ROC Curves for stage 1 in Model 2 for the different algorithms: binary classification.

In the second stage of Model 2, a multi-classification, similar to the one in Model 1, is applied. As the aim of this stage is to refine the diabetic classification, only thermograms of diabetic individuals are used and the same three classes used in Model 1 are considered.

The evaluation of each classification algorithm is measured, separately, for each one of the three classes, using a one-vs.-all method. The results of performance obtained for each class with the different algorithms are shown in Table 5. The comparison of the algorithms is based on the average values, of the three classes, for each of the metrics; these values are shown in the *Average* row of Table 5.

**Table 5.** Metrics scores for stage 2 in Model 2—multi-classification.

| Algorithm | Class | Accuracy | Sensitivity | Specificity | Precision | AUC | F-Score |
|---|---|---|---|---|---|---|---|
| SVM Linear | 1 | 0.964 | 0.946 | 0.975 | 0.959 | 0.996 | 0.952 |
| | 2 | 0.939 | 0.917 | 0.946 | 0.846 | 0.979 | 0.880 |
| | 3 | 0.975 | 0.946 | 0.992 | 0.986 | 0.983 | 0.966 |
| | **Average** | 0.959 | 0.936 | 0.971 | 0.930 | 0.986 | 0.933 |
| SVM Quadratic | 1 | 0.980 | 0.986 | 0.975 | 0.961 | 0.998 | 0.973 |
| | 2 | 0.949 | 0.875 | 0.973 | 0.913 | 0.983 | 0.894 |
| | 3 | 0.969 | 0.959 | 0.975 | 0.959 | 0.981 | 0.959 |
| | **Average** | 0.966 | 0.940 | 0.975 | 0.944 | 0.987 | 0.942 |
| 3-NN | 1 | 0.959 | 0.946 | 0.967 | 0.946 | 0.981 | 0.946 |
| | 2 | 0.918 | 0.833 | 0.946 | 0.833 | 0.940 | 0.833 |
| | 3 | 0.959 | 0.946 | 0.967 | 0.946 | 0.984 | 0.946 |
| | **Average** | 0.946 | 0.908 | 0.960 | 0.908 | 0.968 | 0.908 |
| Weighted k-NN | 1 | 0.964 | 0.946 | 0.975 | 0.959 | 0.996 | 0.952 |
| | 2 | 0.939 | 0.875 | 0.959 | 0.875 | 0.981 | 0.875 |
| | 3 | 0.974 | 0.973 | 0.975 | 0.960 | 0.989 | 0.966 |
| | **Average** | 0.959 | 0.931 | 0.970 | 0.931 | 0.989 | 0.931 |
| LR | 1 | 0.954 | 0.919 | 0.975 | 0.958 | 0.991 | 0.938 |
| | 2 | 0.918 | 0.896 | 0.926 | 0.796 | 0.953 | 0.843 |
| | 3 | 0.964 | 0.932 | 0.984 | 0.972 | 0.982 | 0.952 |
| | **Average** | 0.946 | 0.916 | 0.962 | 0.909 | 0.975 | 0.911 |

Analyzing all the results obtained with the different algorithms, presented in Table 5, and focusing, in particular, on the *Average* row, both SVM Linear and Weighted k-NN algorithms present similar performance, obtaining a difference lower than 1% between each metric, and the same happens with the LR and 3-NN algorithms; however, the first group performs slightly better than the second group. The Quadratic SVM is the algorithm that presents the best results, obtaining in some metrics a difference of 3% when compared to the algorithm that presents worse metric results. In Figure 7, the ROC curves obtained with the SVM Quadratic algorithm, for each one of the three class, are illustrated.

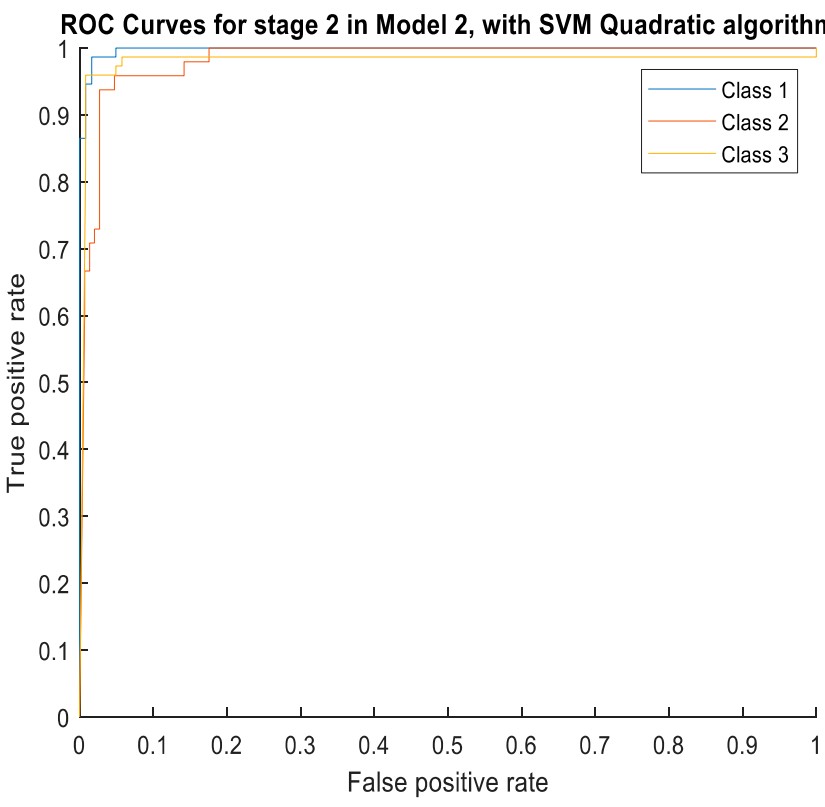

**Figure 7.** ROC curves for stage 2 in Model 2, with SVM Quadratic algorithm.

From Figure 7 it is perceptible that, although all the AUC results are higher than 91%, Class 1 is the one whose performance excels, with the highest AUC value. Classes 2 and 3 present very similar AUC values, but considering the values of other metrics, the results in Class 3 are slightly better than in Class 2.

### 3.4. Comparison of Model 1 with Model 2

In order to compare the performance of both models, the remaining 20% of the dataset not used to train and validate the models was used.

For this comparison, the classification algorithms that showed best performance for each one of the two models were used: as shown in Sections 3.2 and 3.3, the SVM Quadratic algorithm for Model 1, while for Model 2 is used the Logistic Regression algorithm, for stage 1 (binary classification), and the SVM Quadratic algorithm for stage 2 (multi classification).

The results of the performance obtained with both models are presented in Table 6. To compare them, as before, the average of the values of each metric, presented in the *Average* row of Table 6.

**Table 6.** Comparison between Model 1 and Model 2.

| Model | Class | Accuracy | Sensitivity | Specificity | Precision | F-Score |
|---|---|---|---|---|---|---|
| Model 1 | Healthy | 0.833 | 0.667 | 0.896 | 0.706 | 0.686 |
| | 1 | 0.803 | 0.611 | 0.875 | 0.647 | 0.629 |
| | 2 | 0.939 | 0.938 | 0.940 | 0.833 | 0.882 |
| | 3 | 0.970 | 0.929 | 0.981 | 0.929 | 0.929 |
| | Average | 0.886 | 0.786 | 0.923 | 0.779 | 0.781 |
| Model 2 | Healthy | 0.924 | 0.833 | 0.958 | 0.882 | 0.857 |
| | 1 | 0.894 | 0.778 | 0.938 | 0.824 | 0.800 |
| | 2 | 0.939 | 0.938 | 0.940 | 0.833 | 0.882 |
| | 3 | 0.970 | 0.929 | 0.981 | 0.929 | 0.929 |
| | Average | 0.932 | 0.869 | 0.954 | 0.867 | 0.867 |

As only 20% of the data were used to perform the comparison between the two models, which results in a low amount of thermograms per class, it appears that the results of Table 6 are lower when compared to those presented in Table 5 (results of stage 2 in Model 2). However, it must be noted that Model 2 is composed of two stages, and the errors in the binary classification of stage 1 harm the global performance of the model.

Nevertheless, Model 2 performs better than Model 1, for all of the metrics evaluated. It can be verified that the values of the metrics for Classes 2 and 3 are the same for both models, while for the Classes 1 and Healthy the performance is better with Model 2. This result suggests that Model 2 is the one that is able to best discriminate between Healthy individuals and those with slight signs of pathology (Class 1). Additionally, on average, F-score is almost 10% higher for Model 2 and Accuracy 5%; thus, this approach outperforms Model 1 and allows for a more accurate performance classification concerning the healthy category and the first class of diabetic individuals.

## 4. Discussion and Conclusions

As previously mentioned, there are research works, reported in the literature, that apply machine learning or deep learning techniques with good performance results (taking into account the presented metrics) to analyze the temperature distribution on the soles of the feet of diabetic and healthy individuals, using thermograms. These works either aim to classify the thermograms into only two classes—(1) normal/healthy and (2) diabetics— or proceed with a more differentiated classification of the diabetic class. The works of Saminathan et al. [13], Khandakar et al. [16] and Isaza and Diaz [18] use techniques that only allow for a binary classification of thermograms (fitting the first case), while the works of Cruz Vega et al. [9] and of Khandakar et al. [16] (that only use part of the database [15], as

they only involve the classification of thermograms of diabetic individuals into five classes) are in the second group, although in [9] only two by two classes comparisons are performed.

The work developed in this paper aimed to obtain a thermogram classification tool that would allow a response with clinical interest, in the sense of being suitable for being used by health professionals in a clinical environment. Thus, it was intended to develop a tool that is prepared to receive a thermogram at the entrance and that gives as output its classification as healthy, mild diabetic, moderate diabetic or severe diabetic. To this end, the 334 thermograms in the public dataset [15] were used to develop two approaches that allow for the classification of foot thermograms into four classes, either directly, using Model 1, or in two phases, with Model 2. Both models use a one-vs.-all method for class comparison and present good performance (given the obtained values of the metrics); therefore, with this work, we think that we have taken a step towards obtaining a tool that can be used in clinical practice to efficiently classify a diabetic foot.

To synthetize what has been conducted, it is observed that, in this article, a classification methodology is proposed, comparing the performance of two different models, and different machine learning algorithms to classify foot thermograms of healthy and diabetic individuals are presented. For the extraction of the features, the concept of cluster was used both to divide the thermograms into different regions and to extract different regional parameters that are used as features to train and test the models used.

The two proposed models proved to be efficient, allowing us to classify a foot thermogram as belonging to a healthy or diabetic individual, with the diabetic ones divided into three classes. It can be concluded that Model 2, which firstly classifies the feet as healthy or diabetic and then obtains a refinement of the diabetic ones in three different classes, outperforms Model 1, which classifies in just a step a thermogram into four classes (healthy and three other classes for diabetics), allowing for a more precise classification regarding the healthy category and the first class of diabetics, corresponding to an early stage of the disease.

The illustrated results show that the proposed methodology can be a useful tool in helping health professionals in the classification of thermograms and in the early detection of the risk of injuries and in the prevention of ulcerations in the feet.

A limitation of this work is the fact that the used public dataset is unbalanced, thus resulting in the low representation of some classes. As future work, we intend to extend the dataset, in order to balance the number of thermograms of healthy individuals with DM and also the number of diabetics in each class. By using balanced datasets with a significant number of samples, classification models can achieve a more accurate prediction. The extension of the dataset will not only imply the inclusion of more detailed clinical information on the subjects in study (which will allow for a better characterization of the obtained thermograms) but will also involve the application of the concept of active thermography. We observe that active thermography consists of temperature measurement in dynamic states as a result of a certain thermal provocation and allows one to obtain more accurate quantitative information regarding the variation of the temperature [24–26]; in our case, external stimulation will be applied by using a cold bag of gel to slightly cool down the soles of the feet, and then a thermographic camera will be used to record the temperature recovery to the initial conditions.

**Author Contributions:** Conceptualization, V.F. and A.T.; methodology, V.F. and P.T.; software, P.T.; validation, V.F., A.T. and P.T.; formal analysis, V.F. and A.T.; investigation, V.F. and A.T.; resources, V.F., A.T. and P.T.; data curation, V.F., A.T. and P.T.; writing—original draft preparation, A.T. and P.T.; writing—review and editing, V.F. and P.T.; visualization, P.T.; supervision, V.F. and A.T.; project administration, not applicable; funding acquisition, not applicable. All authors have read and agreed to the published version of the manuscript.

**Funding:** This research received no external funding.

**Institutional Review Board Statement:** Not applicable.

**Informed Consent Statement:** Not applicable.

**Data Availability Statement:** The used dataset is public [15] and is available at https://ieee-dataport.org/open-access/plantar-thermogram-database-study-diabetic-foot-complications (accessed on 2 January 2022).

**Acknowledgments:** The research of the Ana Teixeira was partially financed by Portuguese Funds through FCT (Fundação para a Ciência e a Tecnologia) within the Projects UIDB/00013/2020 and UIDP/00013/2020.

**Conflicts of Interest:** The authors declare no conflict of interest.

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
