# Peer review of "Automatic Classification of Foot Thermograms Using Machine Learning Techniques"

_algorithms, doi:10.3390/a15070236_

Round 1

Reviewer 1 Report

The paper proposes a method for automatic classification of foot thermograms using machine learning approach to diagnose a diabetes. The topic is important and interesting. However, a number of issues should be addressed before the paper will be suitable for publication.

1.     Literature review should be performed to show how this problem was addressed by the other researchers. Also, Authors should refer the active thermography which is recently gaining much attention as it provides more reliable results when compared to the classic one. See e.g.:

O’Mahony, et al., Investigation of reconstructed three-dimensional active infrared thermography of buried defects: multiphysics finite elements modelling investigation with initial experimental validation, Journal of Thermal Analysis and Calorimetry, 2020, 142(1), pp. 473-481

Usamentiaga, et al., Infrared thermography for temperature measurement and non-destructive testing, Sensors, 2014, 14(7), pp. 12305-12348

Ruminski, Distributed processing in medical, parametric imaging, Proceedings of the 5th IASTED International Conference on Visualization, Imaging, and Image Processing, 2005, pp. 487-492

Strakowska, et al., A three layer model for the thermal impedance of the human skin: Modeling and experimental measurements, Journal of Mechanics in Medicine and Biology, 2015, 15(4)

Strakowska, et al., Thermal modelling and screening method for skin pathologies using active thermography, Biocybernetics and Biomedical Engineering, 2018, 38(3), pp. 602–610

Nowakowski, et al., Medical applications of model based dynamic thermography, Proceedings of SPIE-The International Society for Optical Engineering, 2001, 4360, pp. 492-503

2.     The details about image acquisition should be provided (a type of camera used, image parameters)

3.     Which features were extracted from thermograms? What feature selection algorithms were used?

4.     How many features where fed to the classifier input? How many samples were classified? How were these classifiers validated?

5.     Obtained results should be compared and discussed with these obtained by other researchers.

6.     Were the results consulted with physicians (diabetologists) to check if the proposed can be applied in the future in the clinical practice?

7.     Limitations of performed study should be provided.

Reviewer 2 Report

1.    Line 252 – lack of reference.
2.    Figure 7 – lines are too thin to see plots. Please make this figure a vector plot (i.e in eps file format)
3.    Why the proposed method is better or how it is different from many, many others classification algorithms using thermograms  that has been developed and published in last three years? Many of them are covered in Introduction section, however there is no deep discussion about that.
4.    Paper “Plantar Thermogram Database for the Study of Diabetic Foot Complications” has 10 reported citations in IEEE and   17 citations in reserchgate and google scholar. Several of those articles are also on same subject as proposed paper (deep learning based classification). Please compare your results with already published methods.
5.    It the proposed method effectiveness limited to the proposed dataset? Has it been evaluated on any other dataset? Is the dataset varied enough to draw general conclusions based on it about method effectiveness?

Round 2

Reviewer 1 Report

Thank you for taking into account all my comments raised in the review. The paper is now suitable for publication.

Reviewer 2 Report

Authors have addressed all my remarks. In my opinion paper can be accepted as it is.